

# How atmospheric CO₂ can inform us on annual and decadal shifts in the biospheric carbon uptake period

Theertha Kariyathan[1,2], Ana Bastos[1], Markus Reichstein[1], Wouter Peters[2,3], and Julia Marshall[4]

[1]Max Planck Institute for Biogeochemistry, Germany
[2]Wageningen University and Research, Environmental Sciences Department, 6708 PB Wageningen, The Netherlands
[3]University of Groningen, Centre for Isotope Research, Groningen, The Netherlands
[4]Deutsches Zentrum für Luft- und Raumfahrt (DLR), Institut für Physik der Atmosphäre, Oberpfaffenhofen, Germany

**Correspondence:** Theertha Kariyathan (tkariya@bgc-jena.mpg.de)

**Abstract.** The carbon uptake period (CUP) refers to the time of each year during which the rate of photosynthetic uptake surpasses that of respiration in the terrestrial biosphere, resulting in a net absorption of $CO_2$ from the atmosphere to the land. Since climate drivers influence both photosynthesis and respiration, the CUP offers valuable insights into how the terrestrial biosphere responds to climate variations and affects the carbon budget. Several studies have assessed large-scale changes in

CUP based on seasonal metrics from $CO_2$ mole fraction measurements. However, an in-depth understanding of the sensitivity of the CUP as derived from the $CO_2$ mole fraction data ($CUP_{MR}$) to actual changes in the CUP of the net ecosystem exchange ($CUP_{NEE}$) is missing. In this study, we specifically assess the impact of (i) atmospheric transport (ii) inter-annual variability in $CUP_{NEE}$ (iii) regional contribution to the signals that integrate at different background sites where $CO_2$ dry air mole fraction measurements are made. We conducted idealized simulations where we imposed known changes ($\Delta$) to the $CUP_{NEE}$ in the

Northern Hemisphere to test the effect of the aforementioned factors in $CUP_{MR}$ metrics at ten Northern Hemisphere sites. Our analysis indicates a significant damping of changes in the simulated $\Delta CUP_{MR}$ due to the integration of signals with varying $CUP_{NEE}$ timing across regions. $CUP_{MR}$ at well-studied sites such as Mauna Loa, Barrow, and Alert showed only 50% of the applied $\Delta CUP_{NEE}$ under non interannually-varying atmospheric transport conditions. Further, our synthetic analyses conclude that interannual variability (IAV) in atmospheric transport accounts for a significant part of the changes in the observed

signals. However, even after separating the contribution of transport IAV, the estimates of surface changes in CUP by previous studies are not likely to provide an accurate magnitude of the actual changes occurring over the surface. The observed signal experiences significant damping as the atmosphere averages out non-synchronous signals from various regions.

## 1 Introduction

Terrestrial ecosystems constitute a net sink of carbon from the atmosphere, mediated by the interplay between photosynthesis

and respiration (autotrophic and heterotrophic). The period between the dates when an ecosystem transitions from being a carbon source to a carbon sink and vice-versa is referred to as the carbon uptake period (CUP) (Gonsamo et al., 2012). During the Northern Hemisphere's CUP, a continuous decline can be observed in atmospheric $CO_2$ mole fraction in many sites across the globe. The CUP as defined by net ecosystem exchange (NEE) will be referred to as $CUP_{NEE}$ and the corresponding period



in the $CO_2$ mole fraction data will be referred to as $CUP_{MR}$. The timing and duration of the $CUP_{NEE}$ and $CUP_{MR}$ are influ-
enced by vegetation phenology and soil respiration, which are in turn influenced by climate variability (Gill et al., 2015; Piao
et al., 2019). For example in northern boreal and temperate ecosystems, warmer temperatures trigger early snowmelt and an
associated early onset of plant growth in spring (Buermann et al., 2018; Zhou et al., 2020). In autumn, warm temperatures lead
to delayed leaf senescence and a longer growing season (Piao et al., 2019; Shen et al., 2022). However, warmer temperatures
can also enhance soil respiration if soil moisture is not limiting, and potentially result in earlier termination of the $CUP_{NEE}$
and $CUP_{MR}$ (Piao et al., 2008). The timing of the $CUP_{MR}$ integrates the signal of ecosystem changes over large spatial scales.
Metrics associated with CUP, e.g. its amplitude have been attributed to Northern Hemisphere greening (e.g. Forkel et al., 2016;
Keeling et al., 1996; Barichivich et al., 2013) and to the intensification of the land carbon sink over the past decades (e.g.
Graven et al., 2013; Ciais et al., 2019).

In previous studies (e.g. Fu et al., 2017, 2019), the $CUP_{NEE}$ has been derived from eddy-covariance measurements of net
$CO_2$ fluxes. However, estimation of the $CUP_{NEE}$ using eddy-covariance flux measurements remains challenging on a global
scale due to the uneven distribution of flux towers over the globe and the small spatial area covered by the footprint of these
towers (Jung et al., 2020; Walther et al., 2022). Therefore, several studies have explored the potential of remote sensing to
estimate the $CUP_{NEE}$ (Churkina et al., 2005; Zhu et al., 2012; Gonsamo et al., 2012). However, while satellite-based indices
provide information about the overall health and activity of vegetation, they cannot distinguish between different components
of the carbon cycle, such as gross primary production (GPP) and ecosystem respiration. In drought-stressed ecosystems, there
may even be periods of carbon release during the growing season (Churkina et al., 2005; Zhu et al., 2012; van der Woude et al.,
2023), influencing $CUP_{NEE}$. The satellite-based indices are closely related to vegetation growth or photosynthesis and charac-
terize the start and end of the growing season (Wang et al., 2022; Zeng et al., 2020), but they do not necessarily capture $CUP_{NEE}$.


Measurements of atmospheric $CO_2$ dry air mole fraction from remote background sites represent the balance between sur-
face emissions and uptake from land and ocean (Keeling et al., 1996) over large spatial scales. The seasonal patterns evident
in these data from the Northern Hemisphere, reflect the terrestrial ecosystem exchange mostly from the high and mid-latitudes
and have been used by previous studies to investigate the changes in the $CUP_{NEE}$ over large spatial scales (e.g. Barichivich
et al., 2012; Piao et al., 2008, 2017). Robust methods were developed for estimation of the CUP from $CO_2$ mixing ratio, such
as the Ensemble of First Derivative (EFD) method from Kariyathan et al. (2023), that was better able to identify changes in the
$CUP_{NEE}$ compared to the conventional use of the dates when the detrended seasonal cycle crossed the zero-value. Even with
refined $CUP_{MR}$ estimation methods, atmospheric transport causes a significant fraction of observed $CO_2$ variations at surface
stations. Inter-annual variations and long-term trends in atmospheric transport can affect the relationship between the seasonal
cycle of atmospheric $CO_2$ observations and surface exchange (Murayama et al., 2007; Piao et al., 2008). Previous studies by
Lintner et al. (2006) suggest a contribution by atmospheric transport to the downward trend in the $CO_2$ seasonal cycle ampli-
tude observed at Mauna Loa (MLO) between 1991 and 2002. Murayama et al. (2007) demonstrated how year-to-year changes
in atmospheric transport create significant inter-annual variations in the downward zero-crossing date of the $CO_2$ seasonal



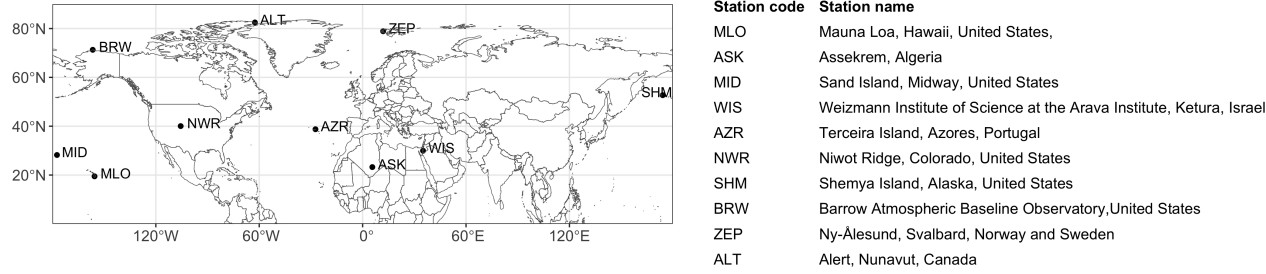

| Station code | Station name |
| --- | --- |
| MLO | Mauna Loa, Hawaii, United States, |
| ASK | Assekrem, Algeria |
| MID | Sand Island, Midway, United States |
| WIS | Weizmann Institute of Science at the Arava Institute, Ketura, Israel |
| AZR | Terceira Island, Azores, Portugal |
| NWR | Niwot Ridge, Colorado, United States |
| SHM | Shemya Island, Alaska, United States |
| BRW | Barrow Atmospheric Baseline Observatory,United States |
| ZEP | Ny-Ålesund, Svalbard, Norway and Sweden |
| ALT | Alert, Nunavut, Canada |

**Figure 1.** Map showing the location of studied sites, with the station names corresponding to the station code shown in the map.

cycle, inevitably influencing $CUP_{MR}$ estimates. Here we aim to understand how well the $CUP_{MR}$ deduced from atmospheric time series observations of $CO_2$ mixing ratios represents the $CUP_{MR}$ changes from the Northern Hemisphere biosphere and its inter-annual variability (IAV), especially:

1 To what extent do $CO_2$ mixing ratio observations accurately capture variations in $CUP_{NEE}$?

2 How does IAV in atmospheric transport affect the observed changes in $CUP_{MR}$?

3 Considering the variability in both $CUP_{NEE}$ and transport, can $CUP_{MR}$ effectively reflect long-term trends in $CUP_{NEE}$?

4 Can the changes observed at the studied sites be attributed to specific regions of the Northern Hemisphere?

To address these questions, we evaluate the role of transport in shaping the $CUP_{MR}$ at regional and global scales, by conducting a series of experiments using the atmospheric transport model TM3 (Heimann and Körner, 2003) for a total of ten sites in the Northern Hemisphere (Fig. 1).

## 2 Methods

To evaluate the degree to which $CUP_{MR}$ represent the changes in the $CUP_{NEE}$, when influenced by atmospheric transport, we design idealized scenarios with prescribed changes to the optimized net ecosystem exchange (NEE) fluxes from the Jena Carboscope Atmospheric $CO_2$ Inversion (Rödenbeck et al., 2003) (version ID: sEXT_ocNEET_v2021). The modifications were applied solely to pixels in the Northern Hemisphere (> 0° N) with a clearly defined seasonal cycle, characterized by a seasonal cycle minimum, and downward and upward zero-crossing points in spring and autumn, respectively. The year 2003 is employed as the reference year (simulations with an alternative reference year, 2001, did not show a noticeable difference), and pixels exhibiting clearly defined seasonal cycles in that specific year were chosen for perturbation. For the remaining pixels, the reference year flux was repeated over time, so that there was no IAV in $CUP_{NEE}$. This was done to ensure that any observed changes in the simulated $CO_2$ mixing ratio could be attributed to the prescribed Δ. The influence of fossil fuel, biomass burning, and





**Table 1.** Description of different forward simulation experiments, using manipulated NEE fluxes. The first character in the experiment name indicates if the early (E) or late (L) $CUP_{NEE}$ phases are manipulated, the next character specifies if Northern Hemisphere (N) or Regional (R) fluxes are adjusted and the subscript and superscript of the last character denote variability (V) in transport and $CUP_{NEE}$ respectively. The $\Delta$ applied in each experiment is shown in the first column. In $\Delta_x^d CUP_{NEE}$, $x$ ranges from -10 to +10 days in intervals of two days. In $\Delta_x^l CUP_{NEE}$, $x$ can be a sequence from -10 to +10 days and vice versa denoted by $p$ and $n$ respectively in the main text.

| $\Delta$ $CUP_{NEE}$ | Period | Spatial Structure | $CUP_{NEE}$ | Transport | Experiment |
|---|---|---|---|---|---|
| | | Northern Hemisphere (NH) | Fixed ($V_0$) | Fixed ($V^0$) | $ENV_0^0$ |
| | Early (E) | | | IAV ($V^T$) | $ENV_0^T$ |
| | | Regional (R) | Fixed | Fixed | $ERV_0^0$ |
| Discrete ($\Delta_x^d$) | | | | IAV | $ERV_0^T$ |
| | | Northern Hemisphere | Fixed | Fixed | $LNV_0^0$ |
| | Late (L) | | | IAV | $LNV_0^T$ |
| | | Regional | Fixed | Fixed | $LRV_0^0$ |
| | | | | IAV | $LRV_0^T$ |
| | Early | Northern Hemisphere | Fixed | IAV | $ENV_0^T$ |
| | Late | Northern Hemisphere | Fixed | IAV | $LNV_0^T$ |
| | | Northern Hemisphere | IAV ($V_1$) | IAV | $ENV_1^T$ |
| | Early | | | | |
| | | Regional | IAV | IAV | $ERV_1^T$ |
| Linear ($\Delta_x^l$) | | Northern Hemisphere | IAV | IAV | $LNV_1^T$ |
| | Late | | | | |
| | | Regional | IAV | IAV | $LRV_1^T$ |
| | Early | Northern Hemisphere | 2 times IAV ($V_2$) | IAV | $ENV_2^T$ |
| | Late | Northern Hemisphere | 2 times IAV | IAV | $LNV_2^T$ |





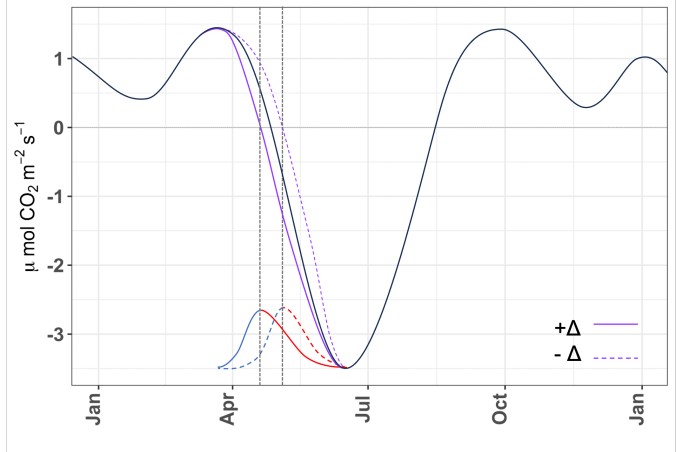

**Figure 2.** Schematic showing manipulation of CUP$_{NEE}$. The shifted purple solid/dashed curves result in +$\Delta$ and -$\Delta$ changes in the CUP$_{NEE}$, respectively. The curve is obtained by subtracting/adding two half-Gaussian curves. For example, the red and blue curves combine at the points (i.e., new onset) indicated by the dashed black lines to produce the purple curves (described in Sect. 2.3). The seasonal cycle minimum separates the early (left) and late (right) CUP$_{NEE}$ phases. The manipulation for the early CUP$_{NEE}$ phase is shown here and can be similarly applied to the late CUP$_{NEE}$ phase.

ocean fluxes on the seasonal variation of atmospheric $CO_2$ is minimal, and changes in the seasonal cycle of atmospheric $CO_2$

reflect alterations in the integrated net ecosystem exchange in the Northern Hemisphere (Barichivich et al., 2012). While these fluxes were not modified in our simulations, our results are based on differences between simulations where only the NEE flux is altered. The flux manipulation was carried out from 1995 to 2017, aligning with the meteorological forcing used in the transport model. These adjusted fluxes were then transported forward using an atmospheric transport model, TM3 (Heimann and Körner, 2003), simulating time series of $CO_2$ mixing ratios at different study sites (as shown in Fig. 1), in temporal fre-

quency aligning with the flask measurements at the sites. After the forward transport run, we assess CUP$_{MR}$ changes based on the simulated $CO_2$ mixing ratios ($\Delta$CUP$_{MR}$) resulting from $\Delta$CUP$_{NEE}$. We use the ensemble of the first derivative (EFD) method from Kariyathan et al. (2023) to evaluate CUP$_{MR}$, as its efficacy on the sites shown in Fig.1 was previously established in Kariyathan et al. (2023).

To evaluate how well CUP$_{MR}$ captures the changes in CUP$_{NEE}$, we used experiment $ENV_0^0$ and $LNV_0^0$, where we imposed spatially uniform, discrete changes in CUP$_{NEE}$ ($\Delta^d$) and the atmospheric transport was held constant in the forward transport run (meaning that one year (2008) of transport was repeated). Then to answer how the IAV in atmospheric transport affects derived CUP$_{MR}$, the $CO_2$ mixing ratios were simulated with inter-annually-varying meteorology (experiment $ENV_0^T$ and $LNV_0^T$). To evaluate the ability of CUP$_{MR}$ to reflect long-term trends in CUP$_{NEE}$, we initially assessed the ability to capture

a trend in CUP$_{NEE}$ while accounting for IAV in atmospheric mixing. This was achieved by prescribing long-term trends in CUP$_{NEE}$($\Delta^l$) and conducting the forward transport run with inter-annually varying meteorology. Subsequently, we then test





the detectability of prescribed linear trends in $\text{CUP}_{\text{NEE}}$ ($\Delta^l$) when IAV was present in both atmospheric transport and NEE (experiments $ENV_1^T$ and $LNV_1^T$). Additionally, to analyze the influence of IAV in $\text{CUP}_{\text{NEE}}$, we prescribed known IAV to $\text{CUP}_{\text{NEE}}$ (experiments $ENV_2^T$ and $LNV_2^T$). Further, to understand the sensitivity of the simulated signals to regional changes 100 (experiments $ERV_0^0$, $LRV_0^0$, $ERV_1^T$ and $LRV_1^T$), we limited the flux manipulation to the different aggregated regions of the TransCom3 (Gurney et al., 2002) experiment. The experiments performed are listed in Table 1.

## 2.1 NEE flux manipulation

The $\text{CUP}_{\text{NEE}}$ is the period when the NEE flux is negative, and the downward and upward zero-crossing dates represent the onset and termination of the $\text{CUP}_{\text{NEE}}$ respectively. Hence, we shift the NEE zero-crossing dates to have a change $\Delta$ (where $\Delta$
is measured in days) in the $\text{CUP}_{\text{NEE}}$ duration ($\Delta\text{CUP}_{\text{NEE}}$). The NEE flux is characterized by daily temporal resolution, showing relatively gradual variations along the y-axis compared to the x-axis. For all the experiments performed, the NEE values (i.e., y-axis) are modified to achieve the desired timing adjustments ($\Delta$) in $\text{CUP}_{\text{NEE}}$, without altering the time axis itself. This adjustment ensures the creation of a smooth curve that closely mirrors the actual flux while achieving the intended change in $\text{CUP}_{\text{NEE}}$. For each value of $\Delta\text{CUP}_{\text{NEE}}$, we modify the downward and upward zero-crossing dates of NEE separately, to evaluate
the effect of changes in the early and late $\text{CUP}_{\text{NEE}}$ phases respectively. This is achieved by adding or subtracting a continuous curve to the period extending from the peak in spring to the NEE minimum for early phase changes and the period from the NEE minimum to peak in winter for late phase changes. The curve is created by combining two distinct half Gaussian curves (Fig. 2, red and blue curves): the first curve has its peak at the new onset/termination and a standard deviation ($\sigma_1$) equal to one-third of the distance between the NEE peak in spring/winter and the new onset/termination. The second curve, also with
its peak at the new onset/termination, has a different standard deviation ($\sigma_2$) equal to one-third of the distance between the new onset/termination and the date corresponding to the NEE minimum value. This configuration (i.e., Gaussian peaks and $\sigma$) ensures that the Gaussian tail minimizes any shift around the NEE peak and trough while realizing the $\Delta$ shift at the onset or termination of the $\text{CUP}_{\text{NEE}}$.

Some pixels exhibit a distinct seasonal pattern without a well-defined peak in spring or winter. In those cases, the period for manipulating the early and late CUP phases then extends from the beginning of the year to the day of minimum NEE and from the day of minimum NEE to the end of the year, respectively. The portions of the first and second curves corresponding to the range from "$\mu$ - $3\sigma_1$" to "$\mu$" and from "$\mu$" to "$\mu$ + $3\sigma_2$," respectively, are then combined and smoothed using a spline function (Fig. 2, purple curves).


We note that the annual flux is not conserved in the manipulation. However, we detrend the simulated $CO_2$ mixing ratio prior to $\text{CUP}_{\text{MR}}$ analysis, which would remove any trend in the $CO_2$ mixing ratio caused by repetition of the manipulated years. Further, when evaluating the simulated $CO_2$ time series, we found that the change in the total annual flux only changes the peak-to-peak amplitude and does not influence the timing and duration of the simulated time series, except at times cor-
responding to periods of manipulation in the $\text{CUP}_{\text{NEE}}$. This happens for instance, when the downward zero crossing of the



NEE flux is manipulated, it changes only the CUP onset and has minimal influence on the CUP termination in the $CO_2$ mixing ratios. The different cases of manipulation are described below:

1. $\Delta_x^d$: In these simulations, every year has the same discrete change in $CUP_{NEE}$. In the different experiments, the magnitude of the shift (denoted by $x$) ranges from -10 to 10 days in intervals of 2 days.

2. $\Delta_x^l$: In these simulations, $\Delta CUP_{NEE}$ progresses from -10 days to +10 days (denoted by $x = p$) or vice versa (denoted by $x = n$) over the period of manipulation.

Manipulation $\Delta_x^d$ is done for experiments where there is no IAV in $CUP_{NEE}$, indicated by $V_0$ in the experiment name. $\Delta_x^l$ manipulation is made for experiments with and without IAV in $CUP_{NEE}$ (i.e., experiment names with $V_0$, $V_1$ and $V_2$). For the case $V_0$, the manipulation is done on the flux of a reference year (chosen arbitrarily 2003), which is repeated in time so that there is no IAV in $CUP_{NEE}$. Any IAV in $CUP_{MR}$ may then be attributed to IAV in transport. In $V_1$, the annual fluxes are used instead of repeating the base year flux. The case $V_2$ has a prescribed IAV in $CUP_{NEE}$. For a given pixel, a set of $\Delta$ values is added to the original $CUP_{NEE}$ in the manipulation period (2000-2017). The set of $\Delta$ has a mean zero and a standard deviation twice that of the IAV in the original $CUP_{NEE}$ for the manipulation period.

The flux alteration is complicated to apply in some cases, as described below :

1. When a local maximum is observed between the downward or upward zero-crossing points and the minimum NEE. In such cases, adding the Gaussian curve shifts these peaks above the zero-crossing line, creating an additional downward or upward zero-crossing point. This complicates the assessment of $CUP_{NEE}$ following the manipulation, and results in $\Delta CUP_{NEE}$ being different from the prescribed value. The $\Delta$ is kept at zero in this case.

2. In a few instances, when the magnitude of $\Delta$ is larger than the period between the original zero-crossing dates and the start/end of the period of manipulation, we instead opt for the next-closest $\Delta$ value in the sequence.

3. Additionally, in manipulation cases where inter-annually varying fluxes are used ($\Delta_x^l$ ), only certain years have the complexities described above. In such instances, the next available $\Delta$ value from the sequence is chosen to minimally impact the imposed $CUP_{NEE}$ trend. This involves selecting a $\Delta$ such that it results in a smaller or larger value compared to the subsequent year, achieving either a positive or negative change in $CUP_{NEE}$ (i.e., $\Delta_p^l$ or $\Delta_n^l CUP_{NEE}$).

Furthermore, the manipulated fluxes are used to conduct regional sensitivity analyses, in which we limit the flux manipulation process explained above to different TransCom3 (Gurney et al., 2002) geographic regions in the Northern Hemisphere. This allows us to evaluate the regional contribution of NEE fluxes to $\Delta CUP_{MR}$ when comparing how perturbations involving different regions are expressed in $\Delta_x^l CUP_{MR}$ at the studied sites. This comparison is conducted for two experiments: $ERV_0^0$, illustrating the integration of signals from various regions in an idealized scenario without IAV in atmospheric transport or $CUP_{NEE}$; and $ERV_1^T$, which reflects signal integration in a relatively realistic setting, with IAV in atmospheric transport and $CUP_{NEE}$.





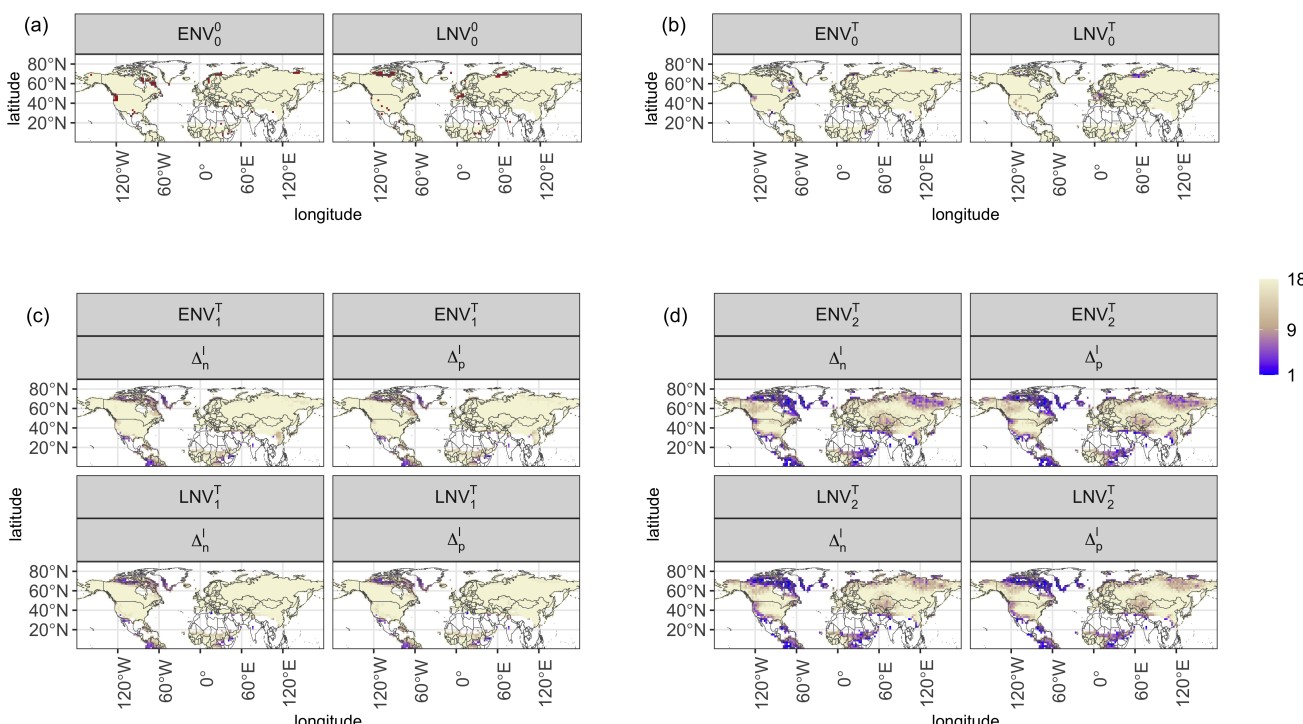

**Figure 3.** Spatial distribution of the pixels manipulated for different experiments: (a) reference year flux is repeated and discrete changes are prescribed to $CUP_{NEE}$ (beige), the red color represents pixels where $\Delta$ is different from the prescribed $\Delta$ due to the complications described in Sect. 2.1. (b) reference year flux is repeated and a long-term trend is applied to $CUP_{NEE}$. When a long-term trend is applied, $\Delta$ varies over the years. The color bar indicates the number of years for which $\Delta$ equals the prescribed $\Delta$ i.e., years with no complications described in Sect. 2.1 (also applicable for panels (c) and (d)). (c) actual $CUP_{NEE}$ is retained and long-term trend is applied to $CUP_{NEE}$. (d) IAV in $CUP_{NEE}$ is doubled and a long-term trend is applied. The panel titles in every plot represent different simulations as detailed in Table 1.





## 2.2 Forward transport runs

We use a three-dimensional global atmospheric transport model, TM3 (Heimann and Körner, 2003), to simulate $CO_2$ mixing ratios at the specified sites based on manipulated NEE fluxes. The model is run at a spatial resolution of 5° in longitude and 4° in latitude with 19 vertical levels, using 6-hourly NCEP reanalysis meteorological fields from 1995 to 2017 and daily surface fluxes from the Jena Carboscope $CO_2$ Inversion (version ID: sEXT_ocNEET_v2021) (Rödenbeck et al., 2003), with the NEE fluxes manipulated as previously described. The forward runs are carried out with 1) fixed transport (meteorology from a random year, here we used the year 2008 and repeated it in time such that there is no IAV) and 2) with inter-annually varying transport for the period 1995 to 2017 to study the contribution of atmospheric transport to the IAV in $CUP_{MR}$. The first five years are excluded from the $CUP_{MR}$ analysis to account for the model's spin-up time, and $\Delta CUP_{NEE}$ is held at zero during this period.

## 2.3 CUP estimation methods

The forward transport runs simulate $CO_2$ mixing ratios at discrete time steps, which we sample at the frequency corresponding to the flask measurements (approximately bi-weekly) at the studied sites. We apply the ensemble of first derivative (EFD) method described in Kariyathan et al. (2023) to the output data and estimate the $CUP_{MR}$. Here, the $CUP_{MR}$ is estimated using a threshold, derived from the first derivative of the detrended and smoothed $CO_2$ mixing ratio seasonal cycle curves. A threshold of 15% and 0% of the first-derivative minimum was used as a threshold to determine the onset and termination of the $CUP_{MR}$, respectively, as in Kariyathan et al. (2023). The calculation is applied to an ensemble of the detrended time series, which allows for an uncertainty range on the CUP estimate to be calculated.

## 3 Results

### 3.1 Northern Hemisphere $CUP_{MR}$ sensitivity under fixed transport

The calculated $\Delta CUP_{MR}$ consistently shows lower absolute values than the prescribed $\Delta CUP_{NEE}$. For example, at BRW, $\Delta CUP_{MR}$ is 0.43 times the prescribed early phase $\Delta CUP_{NEE}$ as illustrated in Fig. 4, (a). This reduction in $\Delta CUP_{MR}$ is found across all the studied sites with varying degrees of intensity as illustrated in Fig. 4, (d), when $\Delta$ is prescribed to either the early or late phases of $CUP_{NEE}$. This shows how atmospheric observations respond differently to CUP perturbations compared to local NEE measurements, and a one-on-one translation might lead to an incorrect interpretation of at least the magnitude of CUP changes. The persistent difference in the magnitude of $\Delta CUP_{MR}$ from the imposed $\Delta CUP_{NEE}$ results from the integration of signals from various regions with different $CUP_{NEE}$ timings as detailed in Sect. 4.

At most studied sites, the $\Delta$ assigned to the early phase of $CUP_{NEE}$ predominantly affects the onset of $CUP_{MR}$ (Fig 4, (b) and (e)). The $\Delta CUP_{MR}$ then corresponds to the changes in onset of $CUP_{MR}$ as indicated by the similar variation in the red bars in Fig. 4 (d) and (e). Similarly, $\Delta$ applied to the late phase of $CUP_{NEE}$ primarily influences the termination of $CUP_{MR}$



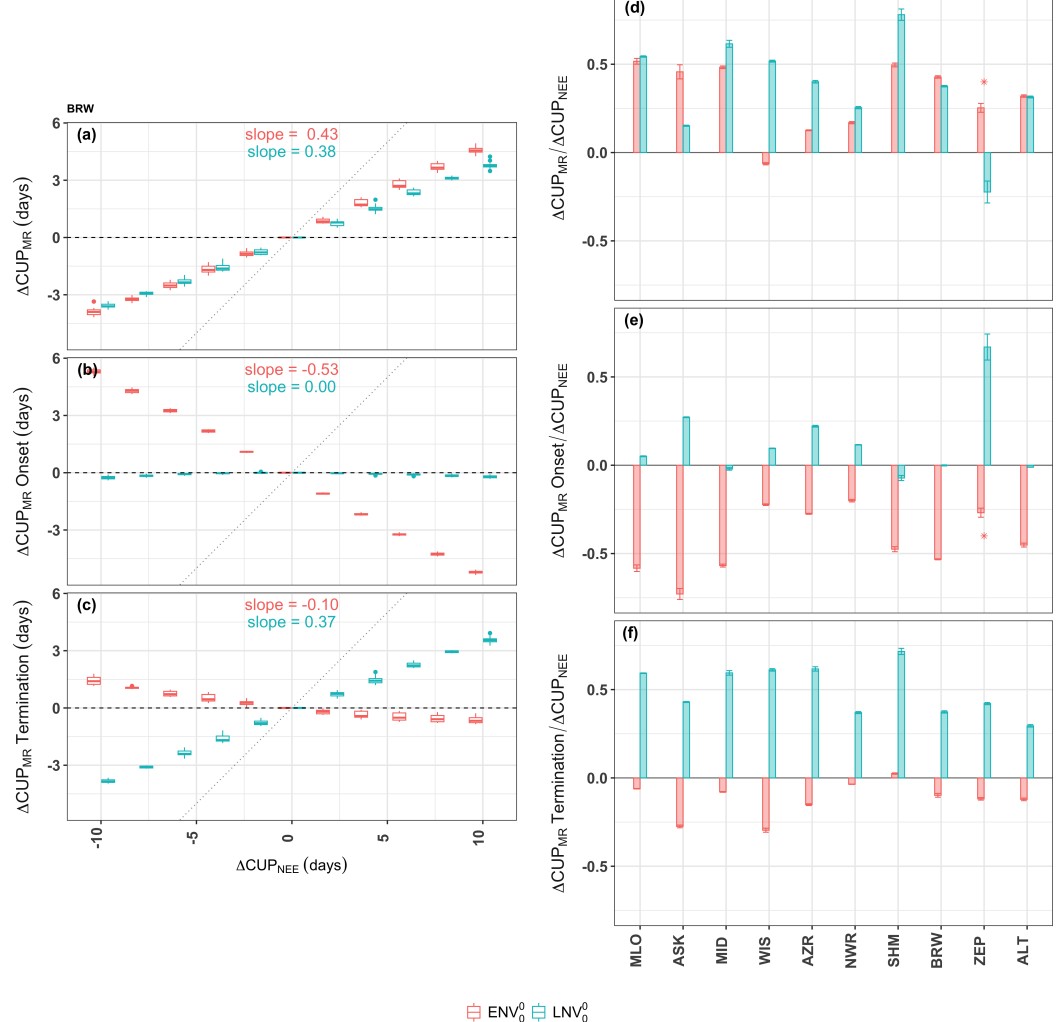

**Figure 4.** The change in CUP$_{\text{MR}}$ metrics in response to varying $\Delta$CUP$_{\text{NEE}}$ for experiments $ENV_0^0$ (red) and $LNV_0^0$ (cyan). The experiments $ENV_0^0$ and $LNV_0^0$ largely drives the $\Delta$ in CUP$_{\text{MR}}$ onset and termination respectively and thereby $\Delta$CUP$_{\text{MR}}$. The left panel shows, the $\Delta$ in (a) CUP$_{\text{MR}}$ (ie., the duration), (b) CUP$_{\text{MR}}$ onset and (c) CUP$_{\text{MR}}$ termination against the applied $\Delta$CUP$_{\text{NEE}}$ for BRW. In these panels, the individual boxplot displays the distribution of the median metrics across years, estimated from the ensemble spread for each year. The distribution of the boxplots against $\Delta$CUP$_{\text{NEE}}$ shows deviation from the one-to-one relation (shown by the dotted line). The text within these plots shows the slope of the regression lines fitted to the median of the boxplots. The right panel ((d) to (f)) shows these slopes (unit less) across the different studied sites. The estimate of ZEP is reduced to 0.1 times the actual value for ease of visualization. Error bars represent $\pm$ one standard deviation ($\sigma$) around the estimated slope.





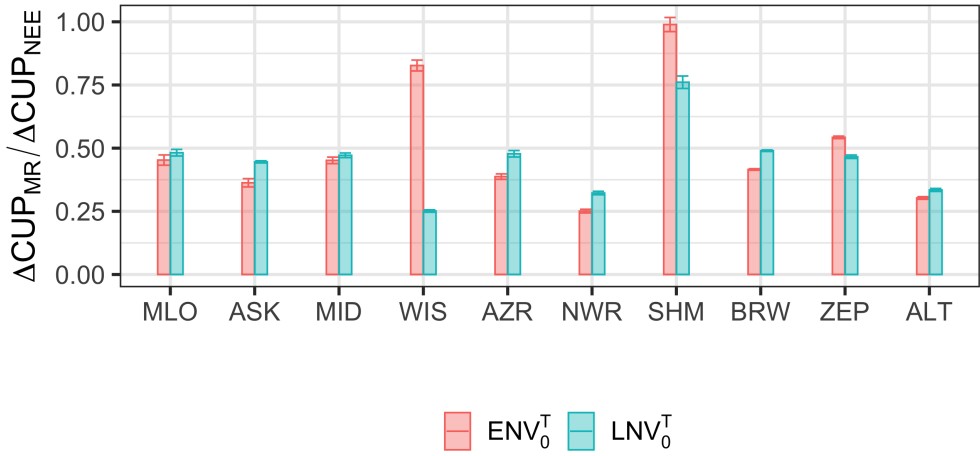

**Figure 5.** The change in $CUP_{MR}$ metrics in response to varying $\Delta CUP_{NEE}$, similar to Fig. 4, (d) but for the experiments with inter-annually varying meteorology, $ENV_0^T$ (red) and $LNV_0^T$ (cyan).

(Fig. 4, (c) and (f)) which then drives $\Delta CUP_{MR}$ in experiment $LNV_0^0$ (Fig. 4, (d) and (f), cyan bars). This suggests that the changes in the early and late phases of CUP at the surface can be analyzed separately by examining the onset and termination

of CUP inferred from $CO_2$ mole fraction observations. Contrary to the direct but dampened relationship between $\Delta CUP_{NEE}$ and $\Delta CUP_{MR}$, we find an opposite response at some sites: a lengthening (shortening) imposed on $CUP_{NEE}$ leads to shortening (lengthening) of the $CUP_{MR}$. This is seen to occur at sites ZEP and WIS, as indicated by the negative slopes at these sites (Fig. 4, (d)).

At ZEP, the late phase $\Delta CUP_{NEE}$ leads to unintended changes in $CUP_{MR}$ onset. For $\Delta$ prescribed to the late $CUP_{NEE}$ phase, the change in $CUP_{MR}$ termination is only 0.4 times the $\Delta$, while that in the onset is 0.6 times the $\Delta$. Thus, the changes intended for $CUP_{MR}$ termination extend to $CUP_{MR}$ onset in the following year. For example, a 10-day delay prescribed to the $CUP_{NEE}$ termination results in a 4-day delay in $CUP_{MR}$ termination and a 6-day delay in the onset. This results in a 2-day shorter $CUP_{MR}$, establishing an inverse relation between $\Delta CUP_{MR}$ and $\Delta CUP_{NEE}$ at ZEP (slope of -0.22 in the experiment $LNV_0^0$).

Likewise, at WIS, in experiment $ENV_0^0$, the change in $CUP_{MR}$ onset is only -0.2 times the applied early phase $\Delta CUP_{NEE}$, while the change in termination is -0.3 times the perturbation imposed. This offsets the $\Delta CUP_{MR}$ and leads to a significant (p < 0.001) inverse relation between $\Delta CUP_{MR}$ and $\Delta CUP_{NEE}$ at WIS (slope of -0.06, in the $ENV_0^0$ experiment).

### 3.2 Northern Hemisphere $CUP_{MR}$ sensitivity under inter-annually varying transport

Even when interannual variations from atmospheric transport are included, changes imposed in $\Delta CUP_{NEE}$ are reflected in

$\Delta CUP_{MR}$. The varying atmospheric transport leads to year-to-year variations in signal integration and changes that were not captured in the experiment with transport from single-year meteorology can be seen in the experiment with inter-annually





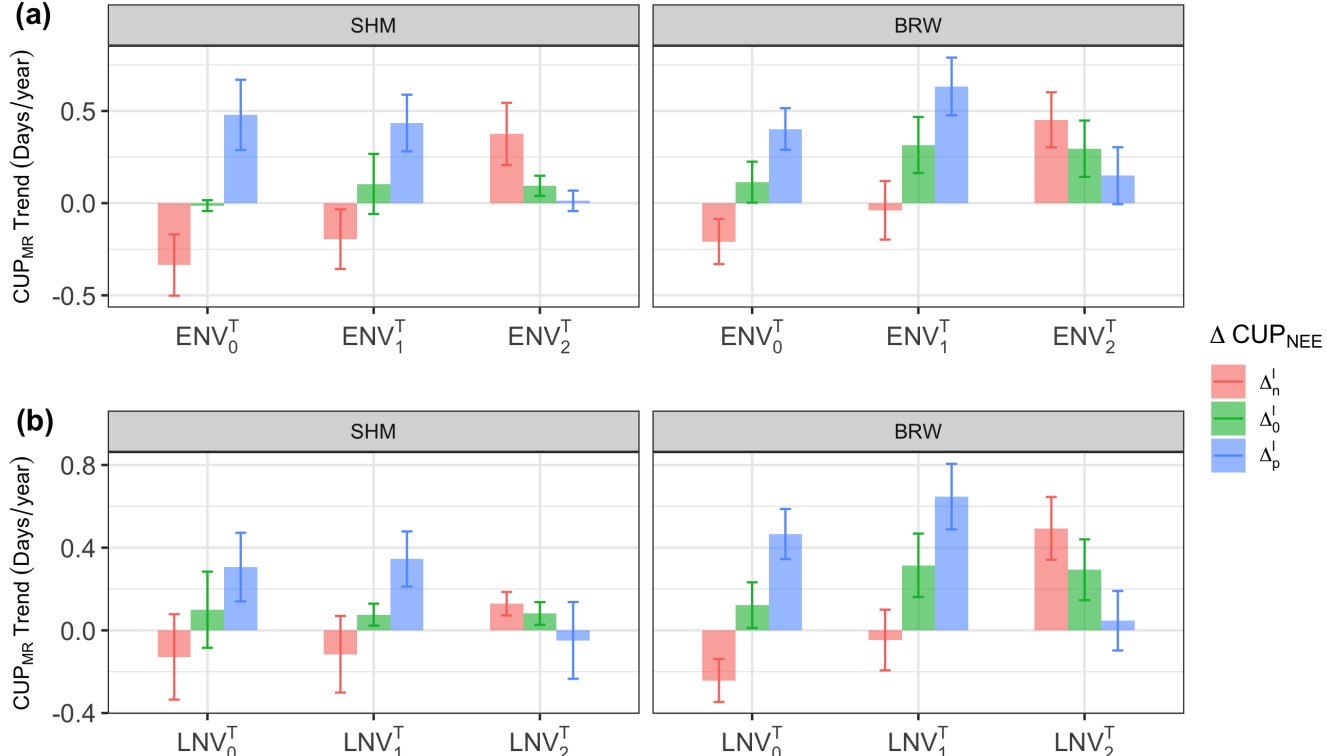

**Figure 6.** Sensitivity of CUP$_{MR}$ to the applied long-term trend in CUP$_{NEE}$ (results for sites SHM and BRW). The bars show the slope of the regression line fitted to the median CUP$_{MR}$ from experiments $ENV_x^T$ (a) and $LNV_x^T$ (b), where $x$ is 0, 1, and 2 implying no IAV in NEE flux, the actual IAV in NEE flux, and two times the actual IAV in NEE flux respectively. Error bars represent $\pm$ one standard deviation ($\sigma$) around the estimated slope. Colours show the prescribed trend $\Delta_p^l$ (1.1 days/year), $\Delta_n^l$ (-1.1 days/year) and $\Delta_0^l$ (0 days/year) applied to CUP$_{NEE}$.

varying transport. This is illustrated for different sites in Fig. 5. An inverse relation between $\Delta$CUP$_{MR}$ and $\Delta$CUP$_{NEE}$ was calculated at WIS and ZEP in experiments $ENV_0^0$ and $LNV_0^0$, respectively, as described in Sect. 3.1. However, in experiments with varying transport, slope values of 0.83 at WIS (experiments $ENV_0^T$) and 0.47 at ZEP (experiment $LNV_0^T$) are found,

compared to -0.06 and -0.22 in the experiment with fixed transport. This suggests that anomalies observed in specific years may be predominantly attributed to the meteorological conditions of those particular years.

### 3.3 Northern Hemisphere CUP$_{MR}$ sensitivity to long-term trends in CUP$_{NEE}$

Out of all the evaluated sites, only SHM and BRW, partially captured CUP$_{MR}$ trends corresponding to the imposed trends in CUP$_{NEE}$ (Fig 6). In experiment $ENV_0^T$, the largest trend in CUP$_{MR}$ is derived at SHM, with values of 0.5 days/year and -0.3

days/year for the imposed increasing (1.11 days/year) and decreasing (-1.11 days/year) trend respectively. Similarly, in experi-



ment $LNV_0^T$, the largest trend in $CUP_{MR}$ is observed at BRW (0.5 days/year and -0.2 days/year for the imposed increasing and decreasing trend respectively). Nevertheless, part of the observed trend can be attributed to the IAV in atmospheric transport thereby showing that the IAV in transport can influence our understanding of the actual long-term changes in $CUP_{NEE}$ trends. This can be seen in Fig. 6, green bars ($\Delta_0^l$), corresponding to experiments $ENV_0^T$ and $LNV_0^T$. Note that in these experiments,

there is no IAV in the $CUP_{NEE}$ flux as indicated by the subscript '0', and $\Delta_0^l$, indicates that no trend is prescribed to the $CUP_{NEE}$. Then any derived $CUP_{MR}$ trend can be attributed solely to the IAV in transport. The trend from the IAV in transport then leads to the asymmetry between the red and blue bars. At BRW, experiment $LNV_0^T$, indicates that a $CUP_{MR}$ trend of 0.1 days/year can arise from variability in transport alone, and accounts for about 20% of the derived $CUP_{MR}$ trend (blue bar showing 0.5 days/year).

Furthermore, we observe that the actual IAV in the $CUP_{NEE}$ fluxes contribute to the derived $CUP_{MR}$ trends, however, as the IAV in the flux becomes larger, it imposes noise that makes the trends harder to detect. This is shown in experiments $ENV_1^T$ and $ENV_2^T$ (Fig. 6), where the actual IAV in $CUP_{NEE}$ is retained and doubled, respectively. In experiment $ENV_1^T$, the $CUP_{MR}$ trend in response to the prescribed opposite trends, $\Delta_p^l$ and $\Delta_n^l$ are distinct in sign. Even though the magnitude of the prescribed trend is the same, a large difference in magnitude can be seen between results for $\Delta_p^l$ (red bar, 0.6 days/year) and $\Delta_p^l$

(blue bar, -0.03 days/year) for example, at BRW. This can be attributed to the increasing trend from both IAV in the actual flux and transport, as indicated by the green bar for $ENV_1^T$. The green bars in experiment $ENV_1^T$ (0.3 days/year) and $ENV_0^T$ (0.1 days/year) are distinct, and their difference (0.2 days/year) is the contribution from the actual IAV in $CUP_{NEE}$ alone. In the experiment where the IAV in $CUP_{NEE}$ per pixel was doubled, it becomes evident that the imposed alterations in $CUP_{NEE}$ are not accurately reflected in $CUP_{MR}$, even at sites like BRW, which exhibited pronounced responses in other experiments

($ENV_1^T$ and $LNV_1^T$).

### 3.4 Regional contribution to $CUP_{MR}$

The various Transcom3 regions of the Northern Hemisphere contribute in various degrees to the $CUP_{MR}$ changes observed at the studied sites. The changes in the Boreal regions are partially captured at both the higher and lower latitudes like ALT, BRW, SHM, MID, and MLO. Considering both early and late phase $\Delta CUP_{NEE}$, the contribution from the Eurasian Boreal region is

largely seen at SHM (-3 days in early and -7 days in the late $CUP_{NEE}$ phase), followed by MID with (-3 in both the days early and late $CUP_{NEE}$ phase), showing an eastward transport from the Eurasian Boreal region. Similarly, considering both the $CUP_{NEE}$ phases, the contribution from the North American Boreal region is seen at all sites except ASK, WIS, and ZEP. In response to delayed onset, prescribed to the $CUP_{NEE}$ in Eurasian Boreal region, a longer $CUP_{MR}$ is calculated at ASK, WIS, and AZR (Fig. 7 (a)), suggesting that the inverse slope relation between $\Delta CUP_{MR}$ and $\Delta CUP_{NEE}$ in Sect. 3.1 might be largely

from changes in the Eurasian Boreal region. In the Eurasian Temperate region, the $\Delta$ prescribed to both the early and late $CUP_{NEE}$ phase, integrate at the lower latitude like MID, MLO, NWR, and, SHM, whereas higher latitude sites like ALT, BRW and ZEP only capture perturbations imposed during the early phase of $CUP_{NEE}$. The contribution from the North American Temperate region is strong during the early phase of $CUP_{NEE}$, while late phase changes are captured only by MID and AZR. Signals from the European region integrate well at most of the studied sites. At ZEP, a significant regional contribution from





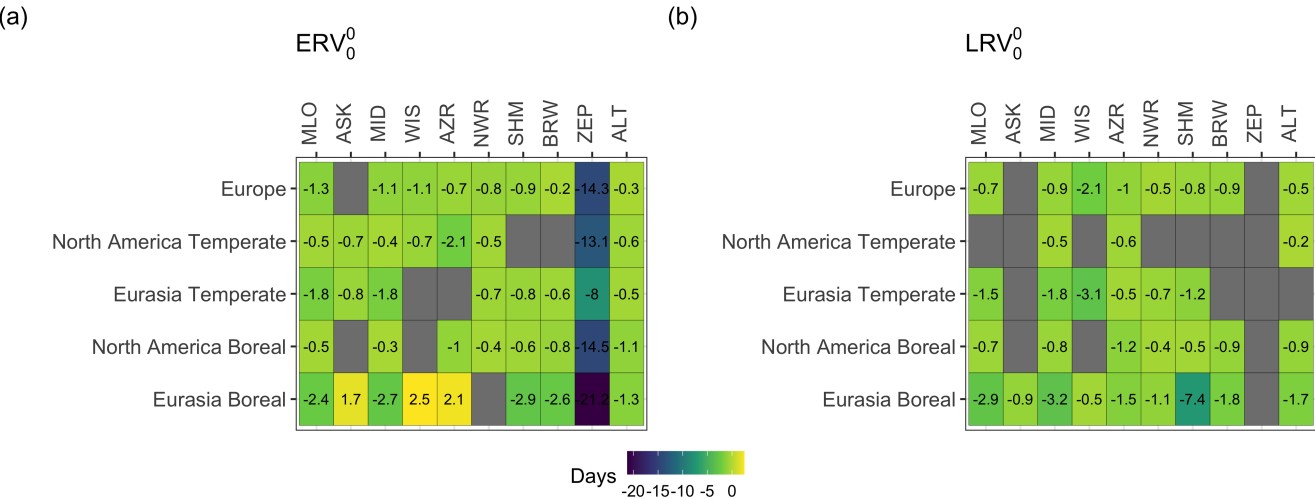

**Figure 7.** Regional contribution to $\Delta CUP_{MR}$. The colour and value represent the ensemble median of $\Delta CUP_{MR}$ when $\Delta CUP_{NEE}$ is -10 days, in experiment $ERV_0^0$ (a) and $LRV_0^0$ (b). Values are displayed solely for the sites where a significant difference in $\Delta CUP_{MR}$ is detected when $\Delta CUP_{NEE}$ is 0 and -10 days (p-value of Mann-Whitney test < 0.05) in the specific region (y-axis).

any of the studied TransCom3 regions can only be seen in the early $CUP_{NEE}$ phase. This explains to some extent the direct relation between $\Delta CUP_{MR}$ and $\Delta CUP_{NEE}$ found only in the early phase (Sect. 3.1).

The long-term trend in the $CUP_{MR}$ could not be accurately attributed to different regions even for sites like BRW that showed a predominant response to the prescribed long-term trend in $CUP_{NEE}$ (Sect. 3.3). This can be seen from (Fig. 8). At BRW, the
$CUP_{MR}$ trends partially reflect the $CUP_{NEE}$ trends prescribed to the Eurasian Boreal region. For example, in the early $CUP_{NEE}$ phase, we find a change of 0.32 days/year and -0.06 days/year in response to the prescribed $\Delta_p^l$ (1.1days/year) and $\Delta_n^l$ (-1.1days/year), respectively. However, the large error bars show that the uncertainty in trend estimation is large when changes are prescribed to only a given TransCom3 region.

## 4   Discussion

We find that changes (both fixed differences and trends) prescribed to $CUP_{NEE}$ are reflected in $CUP_{MR}$ simulated by TM3. However, the magnitude of the change seen in $CUP_{MR}$ is consistently lower than the prescribed change in $CUP_{NEE}$, for example at BRW only about 50% of the change applied to $CUP_{NEE}$ was reflected in $\Delta CUP_{MR}$, even in simulations with fixed transport. This is contradictory to previous studies that consider the long-term $CO_2$ record to reflect changes in surface fluxes. For example, in Piao et al. (2008), 50% of the observed zero-crossing date variance at BRW could be accounted for by NEE
variability.





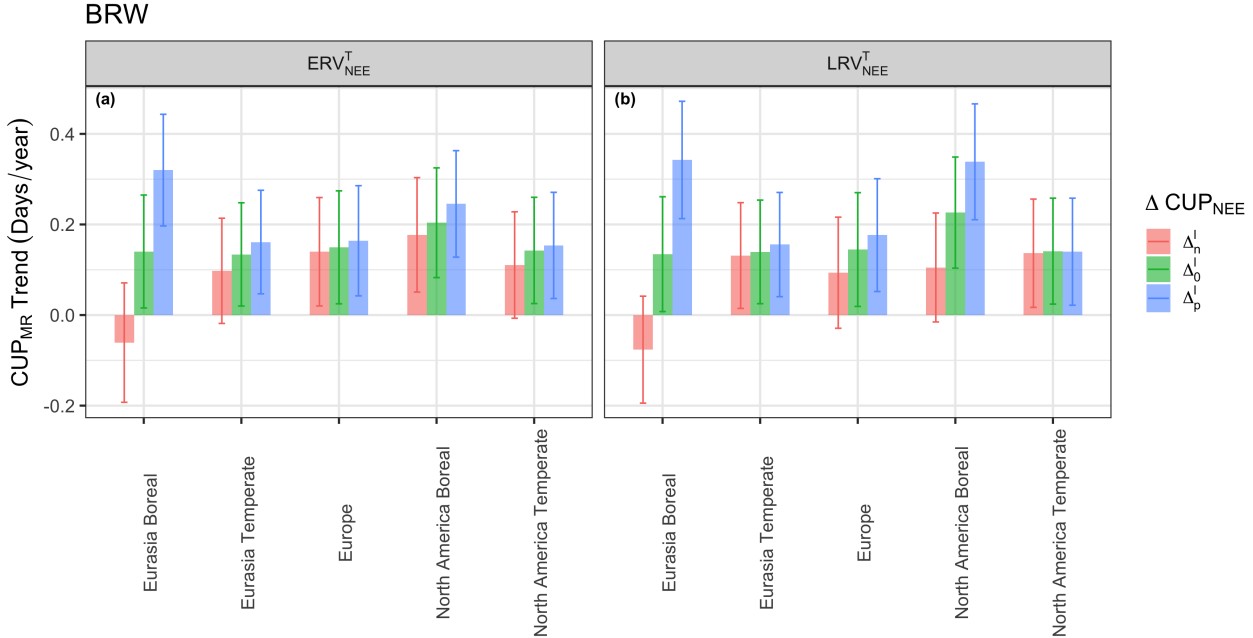

**Figure 8.** Regional contribution to $CUP_{MR}$ trend detected at BRW in response to imposed long-term $CUP_{NEE}$ trends. The bars show the slope of the regression line fitted to median $CUP_{MR}$ from experiments $ERV_1^T$ (a) and $LRV_1^T$ (b). Error bars represent $\pm$ one standard deviation ($\sigma$) around the estimated slope.

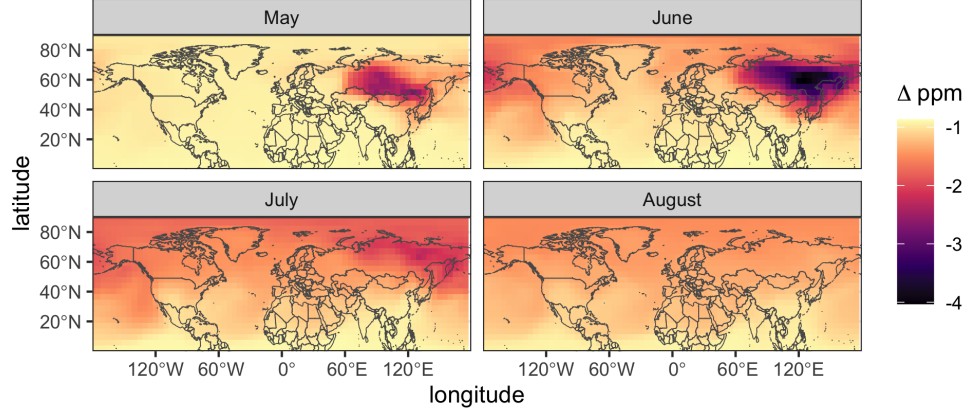

**Figure 9.** 2D mixing ratio fields (integrated vertically up to an altitude of 400 m and averaged per month) for the difference between $\Delta CUP_{NEE}$ of -10 days and 0 days in the experiment $ENV_0^0$, in which $CUP_{NEE}$ from the Eurasian Boreal region was manipulated.





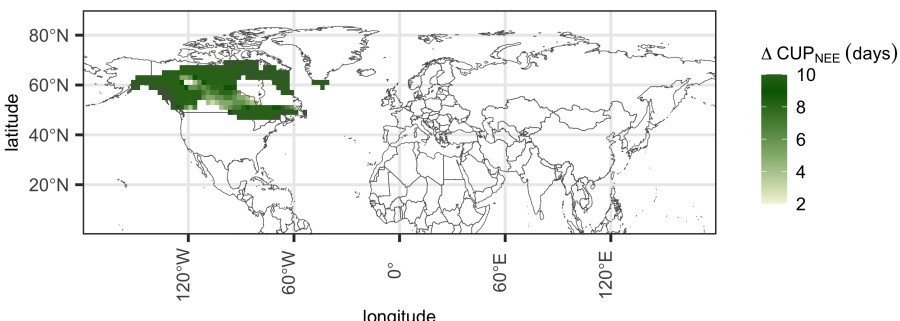

**Figure 10.** The $\Delta CUP_{NEE}$ variability observed across North American Boreal region over a given time period. A significant part of the region, 54% (colored pixels), has $CUP_{NEE}$ onset between days 110 and 150. The colour intensity shows the effective $\Delta CUP_{NEE}$ during this period (days 110-150) for an applied 10-day advancement in $CUP_{NEE}$ onset. For example, pixels with $CUP_{NEE}$ onset on day 118 will see only a $\Delta CUP_{NEE}$ of 8 days for a prescribed 10-day advancement in $CUP_{NEE}$ onset. The color intensity shows such difference in $\Delta CUP_{NEE}$ across the region.

We show that given fixed transport, the reduced expression of changes in $CUP_{MR}$ relative to $CUP_{NEE}$ arises from variations in the timing of $CUP_{NEE}$ across the different regions from where the signal is integrated. For instance, when a delay (10 days) was applied to the $CUP_{NEE}$ onset in the Eurasian Boreal region, the mixing ratio reflects this change over the region in May, and

slowly propagates eastwards by June (Fig. 9), showing a difference in timing of the onset within the Eurasian Boreal region. Due to the difference in $CUP_{NEE}$ timing across the pixels, the "true" $\Delta CUP_{NEE}$ of a region for a period will be different from the applied $\Delta$, as shown in Fig. 10. In a significant portion (54% pixels) of the North American Boreal region (Fig. 10), the onset of $CUP_{NEE}$ typically occurs between days 110 and 150. If $CUP_{NEE}$ fluxes are integrated during this period, the observed signal will represent a substantial portion of the $CUP_{NEE}$ onset from the region. However, when a 10-day advancement in

$CUP_{NEE}$ onset is applied, the true $\Delta CUP_{NEE}$ of the region during this period is only 8.9 days (average of $\Delta CUP_{NEE}$ over the region shown in Fig. 10) and the $CUP_{MR}$ change at the observation site would be less than the applied 10-day $\Delta$. Below, we discuss how the sensitivity of $CUP_{MR}$ to the discrete and long-term changes in surface fluxes is affected when influenced by the interannual variability in both transport and surface fluxes.

## 4.1    Transport influence on $CUP_{MR}$

We have shown the significant role of inter-annually varying atmospheric transport in the evaluation of metrics derived from $CO_2$ mole fraction data. At certain sites such as ZEP and WIS, the $CUP_{MR}$ from simulations with fixed transport failed to capture the $CUP_{NEE}$ changes, whereas in simulations with varying transport, the prescribed $CUP_{NEE}$ changes could be partially derived from $CUP_{MR}$. This indicates that in a given year of meteorology used in the fixed transport simulation, the atmospheric





transport is unlikely to originate from the areas where the $\Delta$CUP$_{NEE}$ was prescribed, while in simulations with transport vari-
ability, the meteorology in other years might have originated from these regions. Thus, the anomalies observed in CUP$_{MR}$ in a
particular year could stem from transport variability rather than anomalies in CUP$_{NEE}$ itself, rendering mixing ratio time-series
less useful for studying inter-annual variations in CUP$_{NEE}$.

Finally, we show that due to the atmospheric transport, the source areas for a given station during the early and late CUP$_{NEE}$
phases can be substantially different, influencing the expression of CUP$_{MR}$ in the different CUP$_{NEE}$ phases. For instance, our
analysis (Fig. 7) shows that WIS mainly receives signals from the Northern Hemisphere land pixels only in the late CUP$_{NEE}$
phase. Consequently, at WIS, the CUP$_{MR}$ is directly proportional solely to changes prescribed to late CUP$_{NEE}$ phase (slope of
0.5 in $LNV_0^0$). Similarly, at ZEP, the contribution of the Northern Hemisphere landmass to $\Delta$CUP$_{MR}$ occurs only in the early
CUP$_{NEE}$ phase. The atmospheric transport to ZEP is dominated by the regions Eurasian Boreal and Europe, during March-May,
in the following months (June-August), the air mass transport is largely over the Northern Atlantic Ocean and does not extend
into the continents in some years (Tunved et al., 2013; Platt et al., 2022). Our observations imply that at ZEP/WIS, changes in
the onset/termination of CUP$_{NEE}$ are more effectively reflected as changes in CUP$_{MR}$.

## 4.2 Long-term trends in CUP$_{MR}$

The long-term trends prescribed to the CUP$_{NEE}$ could be partially derived from CUP$_{MR}$ at BRW and SHM, even under IAV
in atmospheric transport and CUP$_{NEE}$. However, when IAV in CUP$_{NEE}$ was doubled, the prescribed trends were not captured.
This suggests that the long-term trends in the observations may be compromised when there is a higher IAV in CUP$_{NEE}$. The
contribution from atmospheric transport exhibited a CUP$_{MR}$ trend of 0.11 days/year at BRW. This finding aligns with a study by
Murayama et al. (2007) where the IAV in transport alone caused a change of 0.16 days per year in the downward zero-crossing
date at BRW for their analysis period between 1979 and 1999.

Considering BRW, in experiment $ENV_0^T$, the difference between the green and blue bars, representing a CUP$_{MR}$ trend of
0.3 days/year (Fig. 6), is solely due to the imposed trend of 1.1 day/year, showing that the surface flux changes is reduced by
a factor of 0.27. At this site, the actual IAV in CUP$_{NEE}$ causes a CUP$_{MR}$ trend of 0.2 days/ year (see Sect. 3.3). Considering
the reducing factor, a change of approximately 0.7 days/year might be occurring in the surface fluxes. With warming, a longer
growing season is observed in the high latitudes (e.g. Park et al. (2016), 2.6 days per decade). A longer growing season does
not necessarily mean an increase in CUP$_{NEE}$ or CUP$_{MR}$ as they are determined by both photosynthesis and respiration. The
existing literature on the CUP$_{NEE}$ changes in the Northern Hemisphere based on CUP$_{MR}$ varies from increasing (Murayama
et al., 2007) to neutral (Barichivich et al., 2012) to decreasing (Piao et al., 2008; Fu et al., 2017). The complications in inter-
preting CUP$_{NEE}$ changes arise mostly when directly assessing the CUP from CO$_2$ mixing ratio. Therefore, CO$_2$ observations
should preferably be interpreted following a formal inverse estimate of the corresponding surface NEE. It is then possible to
account for the inter-annual variability, trends and delays imposed by the slow atmospheric mixing.



## 4.3 Regional contribution to $CUP_{MR}$

The regions contributing to the integrated signal, at various observation sites are influenced by atmospheric transport to these locations. From the idealized simulations with no IAV in transport nor $CUP_{NEE}$ it turned out that at sites like ALT, BRW, SHM, and MLO, a significant contribution from Boreal and Temperate regions could be calculated, indicating that these remote sites receive well-mixed signals from higher and mid-latitude regions in the Northern Hemisphere with strong seasonality. We calculate that the contribution from mid-latitude is significant at the sites in the Boreal region (e.g. ALT and BRW in early $CUP_{NEE}$ phase) in line with Barnes et al. (2016). They found that the seasonal cycle observed at higher latitude sites is most sensitive to changes in the seasonality of mid-latitude surface emissions, however, we do not find that the mid-latitude influence is more than the Boreal region influences at higher latitude sites. At ZEP, strong signals from both Eurasian and North American regions dominate during the early $CUP_{NEE}$ phase (Fig. 7), and an amplification of the $CUP_{MR}$ signal is found. Further, a significant change in the regional contribution is found between the early and late $CUP_{NEE}$ phases, shifting from continental in the early $CUP_{NEE}$ phase to ocean signals in parts of the late $CUP_{NEE}$ phase (Sect. 4.1). This change explains the significant difference in $\Delta CUP_{MR}$ to $\Delta CUP_{NEE}$ during different phases, as shown in Fig. 4. At WIS, ASK, and AZR, when a delayed onset is imposed on the $CUP_{NEE}$ from Eurasian Boreal regions, a positive $\Delta CUP_{MR}$ (i.e., an extension in $CUP_{MR}$) is calculated. These sites are located in the temperate regions. The $CO_2$ 2D mixing ratio fields (Fig. 9) reveal that changes imposed on the Boreal region propagate partially to the lower latitudes (around 30 degrees north) later in the $CUP_{NEE}$ phase. Thereby the delay imposed on the $CUP_{NEE}$ of the Eurasian Boreal region in May integrates at the lower latitude region (near to the location of WIS, ASK, and AZR) only later in July, delaying and extending the $CUP_{MR}$.

When the long-term trends were applied to specific regions, the slope estimated from $CUP_{MR}$ had large uncertainty due to the influence of IAV in transport and $CUP_{NEE}$. Furthermore, the flux manipulation strictly within the boundaries of the TransCom3 region in our experiments may have substantially limited the regions from which the signals reach the sites. In the real world, regional boundaries are more diffuse, and the footprint of the site provides a more accurate estimate of the regions contributing to the observed signals. Nonetheless, this aspect falls outside the scope of the present study.

The changes in the $CO_2$ mixing ratio time series from the Northern Hemisphere give a larger spatial perspective of the $CUP_{NEE}$ changes. However, results from idealized simulations suggest that they are influenced by atmospheric transport IAV, seasonal changes in atmospheric transport, and IAV in the biospheric fluxes. We find a significant damping of the changes that were imposed on the $CUP_{NEE}$, from the integration of signals from different regions that have varied timing and suggest a more intense change in the local spatial scales. With the constraints in NEE flux manipulation, imposed by the presence of local maxima and insufficient data points for $\Delta$ changes in the early and late $CUP_{NEE}$ (Sect. 2.1), the simulations in this study do not accurately represent the real-world scenarios. In the real world, the changes in $CUP_{NEE}$ are asynchronous across space. Although we broadly examined the influence of different TransCom3 regions, conducting more dedicated footprint analyses of the studied sites may offer further insights into the signals studied here.



## 5 Conclusions

Our analysis reveals that at well-studied sites such as MLO, BRW, and ALT, only circa 50% of the prescribed changes in the $CUP_{NEE}$ fluxes were reflected in $CUP_{MR}$. In simulations with inter-annually varying meteorology, the signals were better captured at a few sites like ZEP and WIS, showing the significant influence of IAV in atmospheric transport. At BRW, 20% of the observed trend could be attributed to the IAV in transport. Furthermore, our findings suggest that the changes estimated in $CUP_{MR}$, subsequent to the separation of atmospheric transport influence, are likely to underestimate the actual magnitude of signals from the surface changes. This is because of the damping due to the integration of asynchronous $CUP_{NEE}$ timing across different regions. While sites like BRW and SHM captured the prescribed long-term changes, they proved insensitive when IAV in $CUP_{NEE}$ was doubled. Furthermore, trends prescribed to individual TransCom3 regions were not captured by the evaluated sites, showing the long-term changes in the seasonal cycle of time series.

*Code and data availability.* The NEE flux used here (Rödenbeck et al., 2003) is available from the Jena CarboScope website at https://www.bgc-jena.mpg.de/CarboScope/?ID=sEXT_ocNEET. The code used for manipulation of the flux is available from the corresponding author on request.

*Author contributions.* The coding and analysis were performed by TK with the contributions of JM. The study was conceptualized by JM, AB, and WP with contributions from MR. The original manuscript was drafted by TK, which was reviewed and edited by AB, WP, JM, and MR.

*Competing interests.* The contact author has declared that none of the authors has any competing interests.

*Acknowledgements.* We thank Christian Rödenbeck for providing access to the NEE flux data from the Jena CarboScope Inversion, as well as for his assistance in resolving queries related to running the TM3 transport model. We acknowledge the assistance of ChatGPT 3.5 for its support in refining the grammatical structure and phrasing of this publication.



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
