# Peer review of "Limitations in the use of atmospheric CO2 observations to directly infer changes in the length of the biospheric carbon uptake period."

_EGUsphere, 2024_

## Community Comment (CC1)

Scientists diverge significantly when defining "concentrations" of gases like $CO_2$ in terms of their atmospheric fractions, and the definition of the term "mixing ratio" varies in more than one regard. Kariyathan et al. (2024) present a representative example where the different definitions are mixed and matched indiscriminately, casting doubt on the results that they present.

The study derives its results using the atmospheric transport model TM3 (Heimann and Körner, 2003), whose authors describe the determinant of tracer transport in terms of "kg tracer mass per kg air mass"; analysis of the units in their Equation (1) makes clear that the air mass must include water vapour. This definition of gas concentration, more simply termed the "mass fraction", is indeed the scalar whose gradients determine diffusive transport (Kowalski et al., 2021). Unfortunately, however, Heimann and Körner (2003) termed this measure the "mixing ratio", which conflicts with both of the popular (yet incompatible) definitions for that term, each referencing *dry* air. And the inclusion or exclusion of water vapour from the definition of this gas concentration is just one of two axes upon which disagreement revolves regarding the meaning of "mixing ratio".

The other axis has to do with whether the fraction is defined on a molar or mass basis. Due to variations in the molecular masses of both air and its components, this distinction is not trivial; for example, although oxygen ($O_2$) is neither extremely heavy nor light, its atmospheric fraction of 20.95% in molar terms (Wallace and Hobbs, 2006) is quite different that in mass terms of 23.15% (Rogers and Yau, 1989). These two measures are affected differently when varying humidity modifies air's effective molecular mass, which is one of many reasons why it is common to eliminate water vapour from the definition of the mixing ratio. But scientists from different disciplines do this in different ways: for meteorologists and other physicists, the mixing ratio is defined as the *mass* of the scalar per unit *mass* of dry air (e.g., Wallace and Hobbs, 2006); by contrast, atmospheric chemists prefer to define the mixing ratio as a *molar* fraction (e.g., Seinfeld and Pandis, 1998), also referencing dry air.

Kariyathan et al. (2024) are at variance with the transport model that they apply regarding both of the two axes described above. Whereas the Heimann and Körner (2003) transport model requires a mass-based fraction that references moist air (including water vapour), the methodology described indicates the use of a molar fraction that references dry air (excluding water vapour).

The methodology refers to the Jena Carboscope Atmospheric $CO_2$ Inversion, citing Rödenbeck et al. (2003), who define $CO_2$ "concentration" (the term "mixing ratio" is not used) in ppm – units that indicate a molar fraction. This Carboscope inversion leans upon the work of Conway et al. (1994) for their $CO_2$ flux database, which defines mixing ratios in ppm (in their Fig. 3), and refers to Komhyr et al. (1983) for $CO_2$ analyses. The latter publication describes a gas flow system that includes a trap to eliminate water vapour prior to analysis of dry air to determine $CO_2$ "concentrations" (in units of ppm; again, the term "mixing ratio" is not used).

All of this history points to a general lack of precision and consistency in defining the $CO_2$ "concentration" whose gradients determine diffusive transport. Kowalski et al. (2021) have shown that differences arising from distinct definitions of $CO_2$ "concentration" – as a mass fraction, molar fraction, or dry mass fraction – cause differences in derived diffusive transport magnitudes that are anything but trivial. And assuming that the requisite atmospheric state data are available, it is a simple accounting exercise to convert the $CO_2$ database to a mass fraction referencing moist air, and thereby feed the TM3 model with the data that it demands.

It would seem worthwhile for Kariyathan and colleagues to indulge in such an exercise, and determine whether their results are sensitive to the precise definition of "mixing ratio", as seems quite likely.

References

Conway, T. J., Tans, P. P., Waterman, L. S., and Kirk, W. T., 1994, Evidence for interannual variability of the carbon cycle from the National Oceanic and Atmospheric Administration/Climate Monitoring and Diagnostics Laboratory Global Air Sampling Network, *Journal of Geophysical Research*, **99** (D11), 22831-22855

Heimann, M. and Körner, S., 2003, The Global Atmospheric Tracer Model TM3: Model Description and User's Manual, Release 3.8a, Max-Planck-Institut für Biogeochemie, Jena, Germany, 131pp.

Kariyathan, T., Bastos, A., Reichstein, M., Peters, W., and Marshall, J., 2024, How atmospheric CO2 can inform us on annual and decadal shifts in the biospheric carbon uptake period, EGUsphere, 2024, 1-22, https://egusphere.copernicus.org/preprints/2024/egusphere-2024-1382, DOI: 10.5194/egusphere-2024-1382

Komhyr, W. D., Waterman, L. S., and Taylor, W. R., 1983, Semiautomatic nondispersive infrared analyzer apparatus for $CO_2$ air sample analyses, *Journal of Geophysical Research*, **88** (C2), 1315-1322.

Kowalski, A. S., Serrano-Ortiz, P., Miranda-García, G., and Fratini, G., 2021, Disentangling turbulent gas diffusion from non-diffusive transport in the boundary layer. *Boundary-Layer Meteorology*, **179** (3), 347-367. https://doi.org/10.1007/s10546-021-00605-5.

Rogers, R. R. and Yau, M. K., 1989, A Short Course in Cloud Physics, 3rd Edition, Elsevier, Burlington, MA, 292pp

Rödenbeck, C., Houweling, S., Gloor, M., and Heimann, M., 2003, $CO_2$ flux history 1982–2001 inferred from atmospheric data using a global inversion of atmospheric transport, *Atmospheric Chemistry and Physics*, **3**, 1919–1964, https://doi.org/10.5194/acp-3-1919-2003.

Seinfeld, J. H. and Pandis, S. N., 2006, Atmospheric Chemistry and Physics: From Air Pollution to Climate Changes, 2nd Edition, John Wiley & Sons, Inc., Hoboken, New Jersey, 1225pp.

Wallace, J. M. and Hobbs, P. V., 2006, Atmospheric Science: An Introductory Survey, 2nd Edition, Elsevier, Amsterdam, 483pp

---

## Author Response (AR1)

Response to the reviews of Preprint egusphere-2024-1382: " How atmospheric $CO_2$ can inform us on annual and decadal shifts in the biospheric carbon uptake period" by Theertha Kariyathan, Ana Bastos, Markus Reichstein, Wouter Peters, and Julia Marshall, to Atmospheric Chemistry and Physics.

Questions from the reviewers are written in **blue**, our answers are in **black**, text copied from the manuscript is written in *italics*, and all changes in the manuscript are typed in ***red italics***. When referencing page, section, and line numbers, we are always referring to the old version of the manuscript.

Answers to Reviewer 1

The authors have conducted a sensitivity analysis of the carbon uptake period (CUP) in the atmospheric $CO_2$ mole fraction to understand how changes in the CUP in net ecosystem exchange (NEE) are reflected in the observed atmospheric mole fraction. Using a series of model simulations, they examined how atmospheric transport and interannual variations in NEE contribute to changes in the CUP in the atmospheric $CO_2$ mole fraction, focusing on 10 observations sites across the northern hemisphere. The authors showed that interannual variations in atmospheric transport provides a significant contribution to changes in the mole fraction CUP. They found that changes in NEE are manifested as damped variations in the $CO_2$ mole fraction, suggesting that the use of observations of changes in the mole fraction CUP to quantify changes in NEE will likely result in an underestimate of the changes in NEE. The authors have developed an interesting approach for assessing the impact of changes in NEE on the atmospheric $CO_2$ mole fraction that would be a valuable contribution to the literature. However, as discussed in my comments below, these results are not surprising. I have some concerns about the observation sites that were selected for the analysis and the resulting conclusions. I would like to see the authors address these concerns before the manuscript can be considered suitable for publication.

We thank you for your thoughtful and constructive comments on our manuscript. We appreciate your recognition of the value of our approach and the potential contribution of our findings to the literature. Below, we address your specific concerns regarding the observation sites selected for the analysis and other comments.

RC1.1

The authors are using 10 observation sites in the northern hemisphere to assess the sensitivity of changes in the $CO_2$ mole fraction to variations in NEE. However, the sites that were selected are those that capture mainly background $CO_2$. As the authors noted, the $CO_2$ model fraction at these background sites "represent the balance between surface emissions and uptake from land and ocean… over large spatial scales," so it is not surprising that these sites will capture a damped manifestation of the changes in NEE. At the end of Section 4.2, the authors stated that "$CO_2$ observations should preferably be interpreted following a formal inverse estimate of the corresponding surface NEE. It is then possible to account for the inter-annual variability, trends and delays imposed by the slow atmospheric mixing." However, inversions using the remote background sites also have difficulties in providing robust estimates of regional changes in NEE

because of the influence of transport and mixing. This is one of the main reasons there has been significant effort focused on expanding the observing network. It is reasonable to ask how would the results of the study change if different sites were selected? For example, would Hegyhatsal (HUN) in Hungary be more sensitive to variations in changes in European NEE? Would East Trout Lake (ETL) in Canada be more sensitive to North American boreal fluxes? There is clearly a need to avoid sites that are strongly sensitive to anthropogenic emissions, but are the 10 sites used in this study the best sites for capturing changes in NEE given the well-known confounding influence of transport and mixing? As a modeling study, it seems to me that the authors have the opportunity to better evaluate the sensitivity of the existing observing network rather than that of what seems to be an arbitrary selection of 10 sites in the network.

Another issue is that the authors are using the temporal frequency of the flask measurements, but that is quite coarse – it is at best weekly. Some sites provide continuous measurements. Is there any benefit to using continuous data for capturing the variations and trends in NEE?

Thank you for the insightful comments. The reviewer first points out that the background sites are expected to capture a dampened response to changes in NEE. Our goal was to quantify the extent of this dampening. For example, well-studied sites like Mauna Loa (MLO), Barrow (BRW), and Alert (ALT) exhibit about a 50% reduction in the observed NEE changes as the atmosphere averages out non-synchronous signals from various regions. Our findings show that the degree of dampening varies based on the site location and the transport dynamics at different times of the year.

The reviewer then notes that it is challenging to get robust estimates of regional NEE changes when using remote background sites for inversions due to the influence of transport and mixing, and an expanded observation network is needed to increase the accuracy of such inversions. Following the comment, we revised line 319:

*Therefore, $CO_2$ observations should preferably be interpreted following a formal inverse estimate of the corresponding surface NEE. It is then possible to account for the inter-annual variability, trends, and delays imposed by the slow atmospheric mixing. Nevertheless, the ability of such inversions to constrain regional changes in NEE can only be improved with an expanded observation network.*

Further, the reviewer asks how results might change with non-background sites like ETL and HUN, their regional sensitivity, and the selection of sites used in this study, while pointing out the importance of minimizing anthropogenic influence.

We agree that the anthropogenic influences should be minimized. To address this, we selected background sites and avoided locations like HUN, which exhibit stronger local influences (Fig. 2). The sites used in this study have long-term data records (Table 1) unlike ETL which starts from 2005 and is not suitable for extracting trends over decades. The ten sites selected for this study represent a chosen subset of the observation network. These sites span the Northern Hemisphere, with roughly at least one station per 10-degree latitude, capturing the spatial diversity of the network. Previous studies, such as Murayama et al. (2007) and Piao et al. (2008), have confirmed that inter-annual variations and long-term trends in atmospheric transport can affect the relationship between the seasonal cycle of atmospheric $CO_2$ observations and surface

exchange. These studies also used a subset of background sites to evaluate transport influence on observed signals, similar to our approach. Additionally, evaluating other sites as suggested by the reviewer, strengthens our conclusion, showing that the current selection adequately reflects how observation sites would capture changes in NEE and the need to avoid non-background sites like HUN. This is discussed further below.

| Station name | Station code | Starting year of the time series |
|---|---|---|
| Mauna Loa, Hawaii, United States | MLO | 1960 |
| Assekrem, Algeria | ASK | 1995 |
| Sand Island, Midway, United States | MID | 1985 |
| Weizmann Institute of Science at the Arava Institute, Ketura, Israel | WIS | 1995 |
| Terceira Island, Azores, Portugal | AZR | 1980 |
| Niwot Ridge, Colorado, United States | NWR | 1968 |
| Shemya Island, Alaska, United States | SHM | 1985 |
| Barrow Atmospheric Baseline Observatory, United States | BRW | 1971 |
| Ny-Ålesund, Svalbard, Norway and Sweden | ZEP | 1994 |
| Alert, Nunavut, Canada | ALT | 1985 |

We acknowledge that different sites are sensitive to various regions. This is one of the challenges we try to address in this study. In section 3.4, we address this by evaluating the regional sensitivity of the studied sites for early and late $CUP_{NEE}$ perturbations. For instance, we find that ALT is more sensitive to changes in North American Boreal NEE fluxes than to changes in Europe NEE fluxes (Figure 7 in manuscript). To address the reviewer's concern regarding the behavior of sites like ETL and HUN within a flux landscape and their regional contributions, we performed additional analyses in which we evaluated $\Delta CUP_{MR}$ at these sites specifically, when $\Delta$ is prescribed to the early phase $CUP_{NEE}$ to the Northern Hemisphere pixels and TransCom3 regions, as outlined in section 2.1.

Figure 1 illustrates $\Delta CUP_{MR}$ for ETL (cyan) and HUN (red) when early-phase $\Delta CUP_{NEE}$ is prescribed to Northern Hemisphere land pixels. The results are consistent with those derived from background sites, with the calculated $\Delta CUP_{MR}$ showing lower absolute values than the prescribed $\Delta CUP_{NEE}$. This can be compared to the red boxes in Figure 4a of the manuscript of BRW, and the slopes in Figure 1 can be compared to the height of the red bars in Figure 4d of the manuscript.

[Figure]

Figure 1: The change in $CUP_{MR}$ in response to varying $\Delta CUP_{NEE}$ for experiment $ENV_0^0$ at ETL (red) and HUN (cyan). In these panels, the individual boxplots display the distribution of the median values across years, estimated from the ensemble spread for each year. The dotted line represents an ideal case of a one-to-one relation between $\Delta CUP_{NEE}$ and $\Delta CUP_{MR}$. The text within these plots shows the slope of the regression lines fitted to the median of the boxplots.

Figure 2 highlights the regional contributions to both sites when $\Delta CUP_{NEE}$ is −10 days. This can be compared to Figure 7a in the manuscript. ETL is primarily influenced by North American boreal fluxes, resulting in a $\Delta CUP_{MR}$ of -1.8 days. HUN is largely influenced by Eurasian boreal fluxes, resulting in a $\Delta CUP_{MR}$ of -2.4 days. Most other regions have a minimal impact on HUN, suggesting that the site is influenced mostly by nearby surface flux changes, which include a larger contribution from anthropogenic influence than the other simulated stations.

[Figure]

Figure 2: Regional contribution to $\Delta CUP_{MR}$ at HUN and ETL. The colour and value represent the ensemble median of $\Delta CUP_{MR}$ when $\Delta CUP_{NEE}$ is -10 days, in experiment $ERV_0^0$. Values are displayed solely for the sites where a significant difference in $\Delta CUP_{MR}$ is detected when $\Delta CUP_{NEE}$ is 0 and -10 days (p-value of Mann-Whitney test < 0.05) in the specific region (y-axis).

The reviewer also mentions that the flask measurements have a relatively coarse temporal resolution, typically on a weekly basis. However, these measurements are sufficient for capturing the larger-scale trends and seasonal variations critical to our analysis. In this study, we focused on flask data as long temporal records are available across a broader network of background sites. When looking into long-term trends, the length of the data record is especially important. Furthermore, it allows for consistent comparison between previous studies that looked into long-term trends. We emphasized this point more clearly in the revised manuscript by modifying line 174 as follows:

*The forward transport runs simulate $CO_2$ mixing ratios at discrete time steps, which we sample at the frequency corresponding to the flask measurements (approximately bi-weekly) at the studied sites. This sampling interval sufficiently captures the larger-scale trends and seasonal variations critical to our analysis and allows for consistent comparison with previous studies that looked into long-term trends using flask measurements.*

RC 1.2.

*The analysis uses the ensemble of the first derivative (EFD) method to estimate the CUP. This approach was described in Kariyathan et al. (2023), but there is no explanation of the method in this manuscript. The reader is forced to go through Kariyathan et al. (2023) to understand what is being done. A brief description of the approach in the current manuscript would be helpful to the reader.*

Thank you for the suggestion. We agree that a brief description of the Ensemble of First Derivative (EFD) method would be helpful for readers who may not be familiar with Kariyathan et al. (2023). We included a concise explanation of the EFD method in the revised manuscript in line 88.

*We use the ensemble of the first derivative (EFD) method from Kariyathan et al. (2023) to evaluate $CUP_{MR}$, as its efficacy on the sites shown in Fig. 1 was previously established in Kariyathan et al. (2023). The method uses an ensemble-based approach to quantify the uncertainty associated with curve-fitting discrete time series data and deriving seasonal cycle metrics. Using this approach, an optimal threshold is defined based on the first derivative of the $CO_2$ seasonal cycle to determine CUP timing. The threshold is selected such that the CUP timing closely corresponds to the spring maximum and late summer minimum, with minimal influence from curve-fitting uncertainty caused by multiple or broader peaks in the $CO_2$ seasonal cycle.*

**Minor comments**
1. Table 1 caption: I believe the subscript refers to the $CUP_{NEE}$, so the caption should read "subscript and superscript of the last character denote $CUP_{NEE}$ and variability (V) in transport, respectively."
   - The reviewer is correct, and the caption has been corrected to read "*the subscript and superscript of the last character denote variability (V) in $CUP_{NEE}$ and transport respectively*".
2. Line 90: Please change: "we used experiment $ENV_0^0$ and LNV" to "we used experiments $ENV_0^0$ and LNV."

- Thank you, this has been corrected.
3. Line 100: Please list the regions over which the fluxes were aggregated.
    - Line 100 has been modified as follows:

    *Further, to understand the sensitivity of the simulated signals to regional changes (experiments $ERV_0^{\ 0}$ , $LRV_0^{\ 0}$ , $ERV_1^{\ T}$ and $LRV_1^{\ T}$ ), we limited the flux manipulation to the 5 Northern Hemisphere land regions of the TransCom3 (Gurney et al., 2002) experiment, namely Europe, Eurasian Temperate, Eurasian Boreal, North American Temperate and North American Boreal.*

4. Lines 137-143: Should Figure 3 be referenced in the text here? I don't believe the figure is actually referenced in the manuscript.

    -Thank you for pointing this out. Figure 3 has now been referenced in line 156

    *The manipulated pixels for the different cases are shown in Fig. 3.*

5. Figure 4 caption: On the second line, change "left panel shows" to "left panels show". On the third line from the bottom, change "right panel ((d) to (f)) shows" to "right panels ((d) to (f)) show"
    - Thanks, the suggested changes have been implemented.

6. Line 218: It would be helpful to see what are the results for the other sites. Can this be presented in a separate table?
    - A table is added to the appendix of the manuscript and referenced in line 218.

    *Out of all the evaluated sites, only SHM and BRW, partially captured $CUP_{MR}$ trends corresponding to the imposed trends in $CUP_{NEE}$ as shown in Fig. 6 ( results for other sites are shown in Table A1).*

7. Lines 226-227: It is unclear that it is the trend from the IAV in transport that "leads to the asymmetry". Perhaps this should be phrased as "the trend from the IAV in transport contributes to the asymmetry."

    -Thank you for the suggestion. Line 226 is rephrased as suggested.

8. Line 300: This description is confusing. Transport from Europe, Russia, and boreal Eurasia is westerly, so it is confusing when the authors say that "air mass transport is largely over the Northern Atlantic Ocean and does not extend into the continents in some years." During summer, because of the warm surface, it is difficult for air transported from Europe and boreal Eurasia to enter the Arctic at low altitudes, so this circumpolar westerly transport occurs at lower latitudes, reducing the sensitivity of the high-latitude ZEP site to Eurasian air masses. I believe that the authors are trying to say that because of the lower latitude circumpolar transport in summer, the sensitivity of the ZEP site to variations in surface $CO_2$ fluxes in summer is confined to the Arctic and does not extend equatorward into the continents.
    - Thank you for mentioning the confusion, we meant that the transport to ZEP is different in the summer months and confined to the Arctic, such that signals from the continent do not extend to the site location (e.g., Tunved et al., 2013). Following the reviewer's suggestion line 300 has been modified:

    *The atmospheric transport to ZEP is dominated by the regions Eurasian Boreal and Europe during March-May, in the following months (June-August), the air*

*mass transport is largely confined to the Arctic and does not extend equatorward into the continents in some years (Tunved et al., 2013; Platt et al., 2022).*

Answers to Reviewer 2

The authors present a modeling study to determine how much surface flux information is contained in the atmospheric $CO_2$ mole fraction observations. They see how well they can recover the time duration when net ecosystem exchange is uptake by the northern hemisphere biosphere from the seasonal cycle of atmospheric $CO_2$ mole fraction observations after running surface fluxes and anomalies through an atmospheric transport model. They show that the surface flux signal information is reduced by mixing in the atmosphere.

We thank the reviewer for summarizing the core focus of our study and highlighting the key aspects of our approach. Below we try to address the concerns and suggestions.

RC2.1

The authors motivate this study by a few published studies of changes in the atmospheric $CO_2$ mole fraction seasonal cycle used to infer northern hemisphere surface flux changes without explicitly considering atmospheric transport. This feels like a strawman. The community already acknowledges the influence of atmospheric transport when linking concentrations to surface fluxes. That is why atmospheric inverse techniques were developed decades ago. Jin et al. (2022) provides an example of a study that uses an atmospheric transport model to separate the influence of surface fluxes and winds on the seasonal cycle at Mauna Loa without using an inverse approach. The references therein identify other studies that also make the point that atmospheric $CO_2$ mole fractions are influenced by fluxes and winds. The authors' most important sentence buried in the middle of the discussion is the main point of this study. "Therefore, $CO_2$ observations should preferably be interpreted following a formal inversion estimate of the corresponding surface NEE. It is then possible to account for the inter-annual variability, trends and delays imposed by the slow atmospheric mixing." I'm not sure what new perspective this study adds.

We thank the reviewer for this thoughtful feedback. We agree that the influence of atmospheric transport on $CO_2$ observations is a well-established topic within the community. In the manuscript, we have acknowledged relevant studies, including Lintner et al. (2006) and Murayama et al. (2007), which utilize transport models to demonstrate this. As per your suggestion, we also include a reference to Jin et al. (2022) in line 55:
*Inter-annual variations and long-term trends in atmospheric transport can affect the relationship between the seasonal cycle of atmospheric $CO_2$ observations and surface exchange (Murayama et al., 2007; Piao et al., 2008). For example, Jin et al. (2022) studied the impact of varying winds and ecological $CO_2$ fluxes on seasonal cycle amplitude trends, finding that shifting winds partially offset the amplitude increase at MLO, contributing nearly 50% to the seasonal cycle amplitude changes between 1959 and 2019. Lintner et al. (2006) suggest a contribution by atmospheric transport to the downward trend in the $CO_2$ seasonal cycle amplitude observed at MLO between 1991 and 2002.*

As rightly noted by the reviewer later in the comments, the novelty of this study lies in its evaluation of the different regional influences on CUP timing. While the role of atmospheric transport in explaining $CO_2$ variations at surface stations is indeed well-known, most previous studies have focused on aspects such as the seasonal cycle's amplitude or zero-crossing times. However, there have been limited studies that thoroughly explored transport influence on CUP timing, particularly with the improved estimation method used here, which offers a closer proxy for $CUP_{NEE}$ (e.g., Barlow et al., 2015). We emphasized this more in the introduction in line 59:

*Previous studies have primarily focused on aspects such as the seasonal cycle amplitude or zero-crossing times. Barlow et al ., 2015 have used the improved CUP estimation method and explored the influence of transport on CUP timing to some extent. In this study, we aim to understand in detail how well the $CUP_{MR}$ deduced from atmospheric time series observations of $CO_2$ mixing ratios represents the $CUP_{MR}$ changes from the Northern Hemisphere biosphere and its inter-annual variability (IAV), especially:*

Our synthetic modeling approach enables us to explore how changes at different spatial scales influence observations across various sites and how information is gradually lost due to flux and transport interannual variability (IAV). Remarkably, even at well-known sites like Mauna Loa, Barrow, and Alert, we found that only about 50% of the uniformly applied $\Delta CUP_{NEE}$ across the entire Northern Hemisphere is detected under non-IAV atmospheric transport and flux conditions. This indicates significant dampening, as the atmosphere mixes non-synchronous signals from different regions. These results suggest that, even when accounting for transport IAV, previous studies may underestimate the actual magnitude of surface-level changes in CUP timing. We consider this a valuable insight that goes beyond earlier studies of changes in CUP, warranting our approach and publication.

RC2.2

Furthermore, the way that the wind influence is described as "the integration of signals with varying $CUP_{NEE}$ timing across regions" and "variations in the timing of $CUP_{NEE}$ across different regions from where the signal is integrated" and in Fig 9 is confusing. They are describing the influence of atmospheric transport or winds, but making it sound like it's a flux synchronization issue.

We thank the reviewer for pointing out the confusion. In Figure 9, the $\Delta CUP_{NEE}$ was prescribed uniformly across the pixels in the Eurasian Boreal region. Despite this, the mixing ratio fields for May and June differ, suggesting a difference in the timing (here onset) of the CUP across pixels. While we address the influence of atmospheric transport in the manuscript, Figure 9 demonstrates that asynchrony in flux timing across pixels also contributes to the dampening observed in the simulated $CUP_{MR}$. To make this clearer we modified lines 271 to 275:

*We show that the reduced expression of changes in $CUP_{MR}$ relative to $CUP_{NEE}$ can also result from the variations in the timing of $CUP_{NEE}$ across the different regions. For instance, when a delay (10 days) was applied to the $CUP_{NEE}$ onset in the Eurasian Boreal region, the mixing ratio first reflects this change over the western part of this region in May. Changes in the eastern part of the domain are seen in June (Fig. 9), showing a difference in the timing of the onset within the Eurasian Boreal region.*

and the figure caption:

*Figure 9. 2D mixing ratio fields (integrated vertically up to an altitude of 400 m and averaged per month) when a delay of 10 days is prescribed to the $CUP_{NEE}$ onset in the Eurasian Boreal region. The field (Δ ppm) is the difference between the 2D mixing ratio fields when $\Delta CUP_{NEE}$ is -10 days and 0 days in experiment $ERV_0^0$.*

RC2.3

Other studies have mostly focused on $CO_2$ seasonal cycle amplitude changes and this study's focus on the duration of the carbon uptake period is a little unique. But this study does hypothetical forward model experiments without using the observed atmospheric $CO_2$ mole fractions to constrain which fluxes anomalies are supported by the observations, if any. What new information about the behavior of the northern hemisphere biosphere is learned here? The authors state "Considering the {transport} reducing factor, a change of approximately 0.7 days/year might be occurring in the surface fluxes."

We appreciate this insightful comment. This study primarily focuses on investigating how climate-driven changes in the timing of carbon uptake and release by the land biosphere are reproduced at different observation sites. To achieve this, we adopt a synthetic modeling approach, which enables us to systematically examine the extent to which various changes would be detectable if they really occurred, as evidence suggests. In the statement, "Considering the {transport} reducing factor, a change of approximately 0.7 days/year might be occurring in the surface fluxes," the term "surface fluxes" does not refer to actual biospheric variability. We acknowledge that the statement is confusing. To ensure clarity and focus on the main message, we removed lines 310 to 314. The primary takeaway from the experiment shown in Fig. 7 is that, with increasing interannual variability in the $CUP_{NEE}$, it becomes increasingly difficult to detect long-term trends from atmospheric time-series data. This may help explain the variability in existing literature (mentioned in line 317) regarding changes in $CUP_{NEE}$ in the Northern Hemisphere.

RC2.4

The nomenclature for the experiments makes it difficult to look at a figure and quickly interpret which experiment combination of fluxes and winds it is comparing. The reader must work very hard to understand and remember the notation. Table 1 helps, but does it have to be that complicated? It seems the paired notation of the Experiment and Delta CUPNEE columns are needed to uniquely identify the tests, especially with ENVTo and LNVTo cases.

We have revised the figures to include more descriptive experiment names in the titles. Specifically, we updated the titles of Figures 4, 5, 6, 7 and 8 as follows:

Figure 4 - **E**arly/**L**ate phase -**N**orthern Hemisphere –No **V**ariabilty $_{CUPNEE}^{Transport}$

Figure 5 - **E**arly/**L**ate phase -**N**orthern Hemisphere – **V**ariabilty$^{Transport}$

Figure 6 - **E**arly/**L**ate phase -**N**orthern Hemisphere – **V**ariabilty $_{CUPNEE}^{Transport}$

Figure 7 - **E**arly/**L**ate phase - **R**egional – **N**o **V**ariabilty $_{CUPNEE}^{Transport}$

Figure 8 - **E**arly/**L**ate phase - **R**egional – **V**ariabilty $_{CUPNEE}^{Transport}$

These changes aim to provide greater clarity and improve the understanding of the experiments presented.

RC2.5

*Misleading title? The finding was that obs can't inform well on surface fluxes, without considering atmospheric transport.*

We thank the reviewer for pointing this out. Our intention was for the title to be interpreted as a question. However, to improve clarity and better reflect the findings, we modified the title to: *"Limitations in the use of atmospheric $CO_2$ observations to directly infer changes in the length of the biospheric carbon uptake period."*

Other changes
Line 4 of the Figure 4 caption is rephrased for clarity as: *"The dotted line represents the ideal one-to-one relationship between $\Delta CUP_{NEE}$ and $\Delta CUP_{MR}$."*
In the caption of Figure 8, Line 3, added : "*The colours represent the trend in NEE imposed in the experiment, as described in Fig. 6.*"

References:

Barlow, J. M., Palmer, P. I., Bruhwiler, L. M., and Tans, P.: Analysis of CO2 mole fraction data: first evidence of large-scale changes in CO2
uptake at high northern latitudes, Atmospheric Chemistry and Physics, 15, 13 739–13 758, https://doi.org/10.5194/acp-15-13739-2015, 2015.

Jin, Y., Keeling, R. F., Rödenbeck, C., Patra, P. K., Piper, S. C., & Schwartzman, A. (2022). Impact of changing winds on the Mauna Loa CO2 seasonal cycle in relation to the Pacific Decadal Oscillation. *Journal of Geophysical Research: Atmospheres*, 127, e2021JD035892. https://doi.org/10.1029/2021JD035892

Murayama, S., Higuchi, K., and Taguchi, S.: Influence of atmospheric transport on the inter-annual variation of the CO2 seasonal cycle 450 downward zero-crossing, Geophysical Research Letters, 34, https://doi.org/10.1029/2006GL028389, 2007.

Piao, S., Ciais, P., Friedlingstein, P., Peylin, P., Reichstein, M., Luyssaert, S., Margolis, H., Fang, J., Barr, A., Chen, A., Grelle, A., Hollinger, 455 D., Laurila, T., Lindroth, A., Richardson, A., and Vesala, T.: Net carbon dioxide losses of northern ecosystems in response to autumn warming, Nature, 451, 49–52, https://doi.org/10.1038/nature06444, 2008.

Tunved, P., Ström, J., and Krejci, R.: Arctic aerosol life cycle: linking aerosol size distributions observed between 2000 and 2010 with 475 air mass transport and precipitation at Zeppelin station, Ny-Ålesund, Svalbard, Atmospheric Chemistry and Physics, 13, 3643–3660, https://doi.org/10.5194/acp-13-3643-2013, 2013.